# Prevention and Control of *Fusarium* spp., the Causal Agents of Onion (*Allium cepa*) Basal Rot

Ofir Degani [1,2,*], Elhanan Dimant [1], Asaf Gordani [1,2], Shaul Graph [1] and Eliyahu Margalit [3]

[1] MIGAL—Galilee Research Institute, Tarshish 2, Kiryat Shmona 1101600, Israel
[2] Faculty of Sciences, Tel-Hai College, Upper Galilee, Tel-Hai 1220800, Israel
[3] Extension Service, Israel Ministry of Agriculture and Rural Development, P.O.B 30, Bayit Dagan 5025001, Israel
[*] Correspondence: d-ofir@bezeqint.net or ofird@telhai.ac.il; Tel.: +972-54-678-0114

**Abstract:** *Fusarium* basal rot disease (FBR) is considered a serious threat to commercial onion production in Israel and worldwide. Today, coping means applied in Israel against the disease have limited efficiency and include a four-year crop cycle and disinfecting the soil with metam sodium. At the same time, agricultural tools (harrows, plows, etc.), contaminated equipment and workers facilitate spread of the disease to new growth areas, and the field disease incidence in Israel now reaches 8% of yields in heavily infected areas. Infected onions do not always show disease symptoms and the problem worsens if they arrive at storage facilities, especially since this pathogen genus produces known toxins. The current study aims at examining the potential of chemical control to reduce the damage caused by this disease. To this end, nine commercial fungicides were scanned in plate sensitivity assay against the main pathogens involved, *Fusarium oxysporum* f. sp. *cepae* and *Fusarium acutatum*. Several fungicides were found to be highly effective against the two pathogens, especially the mixtures Azoxystrobin + Difenoconazole, Fluopyram + Trifloxystrobin, or the Fluazinam compounds. Three selected preparations previously tested in seedlings were evaluated here in a full growing season. Prochloraz successfully protected the Orlando variety (white onion, Riverside cv.) and the Noam variety (red onion) at all growth stages against *F. oxysporum* f. sp. *cepae*. At the same time, this treatment was ineffective against *F. acutatum* in Noam cv. Another anti-fungal preparation, Fludioxonil + Sedaxen mixture, showed a wider range of effectiveness at the season's end against the two *Fusarium* species tested in both onion cultivars. These results are an important step towards developing FBR control in commercial onion fields. Follow-up work is needed to optimize the pesticides' concentrations and their application methods and to test them on a field scale. Interestingly, these pathogens were more aggressive towards the cultivar from which they were isolated: *F. oxysporum* f. sp. *cepae* to the red onion Noam cv. and *F. acutatum* to the white Orlando cv. Infecting the plants with both pathogens reduced disease symptoms in the white Orlando cv, suggesting antagonistic interactions in this onion genotype.

**Keywords:** *Allium cepa*; basal rot; chemical control; fungus; *Fusarium*; onion; pathogenicity assay; Prochloraz

## 1. Introduction

*Allium cepa* (the common onion or bulb onion) is one of the most common and consumed vegetable crops worldwide. The total area of dry onion cultivation in Israel is 3523 ha, and the volume production is 76,378 tons (FAOSTAT, 2020 Food and Agriculture Commodity Production data). This cultivation consists of different varieties according to the preferred growing dates [1]. Yet, the leading summer onion, Orlando (Riverside cv., white onion) makes up about 45% of the growing area of dry onion and is grown in all parts of the country. In recent years, new reports have accumulated from farmers about an increase in cases and the spread of *Fusarium* onion basal rot disease (FBR) in Israel's

onion fields, with new reports in the north of the country [2]. The disease is characterized by massive damage to the roots, disrupting their development and causing rotting that spreads upwards to the onion's basal plate and bulbs. Losses from FBR are reported by onion growers around the globe [3]. The disease is caused by species of the *Fusarium* genus that specialize in this host's plants (formae speciales *cepa*). The most common pathogen is *Fusarium oxysporum* f. sp. *cepae* [4], which can cause more than 50% yield losses [5]. *Fusarium* species include notorious pathogens having a global impact [6]. They spread and survive long periods in the soil and can specialize in over 120 plant host species [7].

Seedlings and dormant- or post-harvest bulbs are the most susceptible to FBR. Indeed the disease is sometimes referred to as the "dieback" or "damping-off" disease of the seedlings, i.e., the disease causes seedling mortality [8]. Still, FBR diseases can occur at all the *Allium* development stages. Symptoms become more noticeable during the plants' maturity. The first symptom of plants being infected is the yellowing of leaves, followed by symptoms of curly and withered leaves. Subsequently, these symptoms begin to spread downward. Rotting of bulbs and the root abscission layer appearance are apparent signs of infection, leading to easily uprooting the plants and bulbs from roots [2].

Onion FBR has been documented in Israel for many years, and crop protection methods are based on pre-sowing soil disinfection with metam sodium and a four-year crop rotation. Such soil treatment is efficient because it effectively reduces pathogenic propagules and is specially designed to control soil-borne diseases [9]. Yet, while the former evaluation estimated field losses at about 1%, new data revealed that the disease's damages worsened more than we had thought and can reach 8% (Ofir Degani personal communication). High temperatures before and during bulb maturation are optimal for fungus infiltration, establishment, and spread in the field [4]; thus, global warming may enhance this trend. Furthermore, disease losses are not limited to the field. Infected onions without external disease symptoms can reach storage facilities. FBR can infect other onion bulbs in open shed storage or packing houses. Moreover, these pathogens produce known toxins [10] that can be present in apparently healthy bulbs shipped to the markets [11].

We had previously sampled rotten onions from fields in the Golan Heights (northeastern Israel) during the summers of 2017 and 2018 [2]. The pathogens were isolated from infected tissues of the onion samples and identified based on the fungus colony morphology and microscopic taxonomic keys. Final confirmation of the identity of the pathogens was made using PCR and sequencing of DNA segments using specific primers for fungi and *Fusarium* spp. Four *Fusarium* species were isolated from the onion samples: *F. proliferatum*, *F. oxysporum* f. sp. *cepae*, and two other species less familiar with the causes of the disease—*F. acutatum* and *F. anthophilium* [2]. Phylogenetic analysis showed that these species are divided into two populations, a group isolated from white onions (Orlando cv.) and a group isolated from red onions (varieties no. 565/505), both from Hazera Seeds Ltd. (Berurim MP Shikmim, Israel).

Pathogenicity tests done with seeds and detached mature onions under humid conditions proved that all species are able to cause the disease symptoms but with different levels of virulence. Inoculating seeds with each of the four species separately under in vitro conditions significantly inhibited germination rate, sprout development, and weight gain. Detached mature onions that were treated with these pathogens showed typical symptoms after 14 days, and the fungi were re-isolated from them to satisfy the Koch postulates.

Among other soil fumigants, FBR control today is based on soil treatments with benomyl, Carbendazim, and Prochloraz as a sole treatment or combined with other pesticides [8,12]. Seed coating and bulb dip treatment in these and other preparations provide promising results (reviewed by [8]). Still, none of the above formulas have been inspected against FBR in Israel, and hence, they are not implemented in commercial fields.

Successful chemical control methods can lose effectiveness or be unfitted to other areas. One central factor is the evolution of fungicide resistance that accelerates when extensive fungicides are applied in the same field for a long duration [13]. Another influencing factor is the variations in the *Fusarium* population composition. Thus, the efficacy of

chemicals in controlling FBR in the field greatly varies and may not provide sufficient disease protection [14]. Although single *Fusarium* species could cause FBR, a complex of *Fusarium* species is frequently found to be responsible for the disease. Indeed, as many as 14 different *Fusarium* species were identified as causal agents of FBR around the globe [8]. Each of these species may respond differently to the chemical treatment, as proved by us lately [15]. Thus, searching for and developing new chemical-based treatments with new compounds is a continued effort with high priority. Essentially, to overcome the potential risk that the fungus will become resistant to fungicides, incorporating two or more active ingredients with a different mode of operation is necessary for long-term use.

We recently evaluated commercial chemical fungicides as control treatments against two primary pathogens involved in the disease in Israel, *F. oxysporum* f. sp. *cepae* and *F. acutatum* [15]. Plates screening of fungicides, in vitro seed pathogenicity assay, and potted seedlings in a growth room revealed that Prochloraz has significant pest control potential against *F. oxysporum* f. sp. *cepae*. disease in Orlando cv. This protective shield evaluated at the plants' first growth stages left an open question: will these treatments be effective over a whole growth period? Moreover, the study results imply that no single defense suit can be efficient against all disease causal agents, and that any solution must be based on a tailored response to each of the pathogens involved and the specific onion strain. Thus, new anti-fungal compounds (sole active ingredients or mixtures) should be inspected. Those that have already shown a protective potential under controlled conditions should be developed further and tested in field-like simulations.

Despite significant progress in recent years in research on the subject, significant knowledge gaps still exist regarding FBR pathogenesis in different cultivars, the interrelationships between the pathogens involved in causing it, the disease spread (in the field and in storage) and protective methods that could be applied. There are currently no FBR-resistant onion varieties in Israel and no use of proven pesticides (chemical or biological) is being made against the pathogens. Similar situations exist in other countries (for example, Turkey [16]). Instead, FBR is controlled by chemically-treated seeds with broad-spectrum Antracol (dithiocarbamates) and Carbendazim (systemic, benzimidazole) [16]. Nevertheless, these fungicides can kill beneficial microorganisms in the soil. Moreover, by using agro-pesticides with a sole target site, resistance can quickly become a severe problem [13]. In India, chemical control of FBR and resistant cultivars have limited effectiveness [17].

The current study is part of this continued effort. It aims at identifying new highly effective anti-fungal compounds and inspecting those previously tested in sprouts over an entire growing season. To this end, we scanned nine commercial fungicides at three dosages against *F. oxysporum* f. sp. *cepae* and *F. acutatum* in an in vitro plate assay. These two isolates were more prevalent in field samples of basal rot diseased onion bulbs [2]. Therefore they were chosen for our previous chemical control study [15] and the current follow-up work. We also examined two onion varieties (white Orlando cv. and red Noam cv.) in semi-field pot experiments in an open-air enclosure. Naturally infected soil from the two cultivars' commercial fields was enriched with the two pathogens and challenged with three chemo-protectants (at two dosages each). The impact of these treatments was evaluated by tracking the plants' aboveground emergence and their development indexes at 65 and 115 days after fertilization (DAS), and by stating the plants' survival rate and degree of dehydration (the shoot water content).

## 2. Materials and Methods

### 2.1. Fungicides Culture Plates Assay

The effect of selected pesticides in inhibiting the *Fusarium* onion basal rot disease (FBR) pathogens, *F. oxysporum* f. sp. *cepae* (B14 strain) and *F. acutatum* (B5 strain), was evaluated in culture plates. Petri dishes (9 mm) containing potato dextrose agar medium (PDA; Difco Laboratories, Detroit, MI, USA) were prepared in which different commercial pesticides (Table 1) were combined with a preparation concentration of 1, 10 and 100 parts per million (ppm). The fungicide concentration range tested in the plate assay is quite

wide. The minimum of 1 ppm concentration often has minor to no impact on the fungal growth in many anti-fungal preparations. On the other side, the high 100 ppm dosage can drastically reduce the fungal colony growth in successful pesticides. Thus, it is an adequate concentration range for these kinds of tests. A similar dosage range was previously used by us ([15,18]) and by others [16,19] against plant fungal pathogens. Each concentration was tested in four repetitions. Additionally, control plates were prepared with a PDA substrate (without the pesticide addition). A fungus mycelium disk (6 mm diameter) was cut from the edge of an 8-day-old colony that grew in the dark at $28 \pm 1\ °C$. The colony agar disk was sown in the center of each fungicide-embedded plate. The plates were incubated at $28 \pm 1\ °C$ in the dark. After six days, the fungus colony diameter in each plate was measured and compared to the control.

**Table 1.** Details of the preparations used in this study [1].

| Fungicide Name and Acronym | Manufacturer Supplier | Active Ingredient (Common Name) | Group Name | Chemical Group | Target Site of Action | AI (g/l) | Test [2] |
|---|---|---|---|---|---|---|---|
| Hosen (Flutr) | Cheminova (Lemvig, Denmark) Makhteshim Agan (Airport City, Israel) | Flutriafol | DMI-fungicides (demethylation inhibitors) | Triazoles | Disrupt C14-demethylation in sterol biosynthesis (erg11/cyp51) | 125 | Plates assay |
| Ortiva top(Az-Di) | Syngenta (Basel, Switzerland) | Azoxystrobin | QoI-fungicides (quinone outside inhibitors) | Methoxy-acrylates | Respiration C3:cytochrome bc1(ubiquinol oxidase) at Qo site (cyt b gene) | 250 | Plates assay |
| | Adama Makhteshim (Airport City, Israel) | Difenoconazole | DMI-fungicides (DeMethylation Inhibitors, SBI: Class I) | Triazoles | Sterol biosynthesis in membranes G1:C14-demethylase in sterol biosynthesis (erg11/cyp51) | 125 | |
| Luna sensation (Fluop-Tr) | Bayer CropScience (Monheim am Rhein, Germany) | Fluopyram (Velum) | SDHI (succinate dehydrogenase inhibitors) | Pyridinyl-ethyl-benzamides | Respiration C2: complex II: succinate-dehydrogenase | 250 | Plates assay |
| | Lidorr Chemicals Ltd. (Ramat Hasharon, Israel) | Trifloxystrobin (Flint) | QoI-fungicides (Quinone outside Inhibitors) | Oximino acetates | Respiration C3: complex III: cytochrome bc1 (ubiquinol oxidase) at Qo site (cyt b gene) | 250 | |
| Skipper (Di) | Syngenta (Basel, Switzerland) Tapazol Chemical Industries Ltd., (Beit Shemesh, Israel) | Difenoconazole | DMI-fungicides (DeMethylation Inhibitors, SBI: Class I) | Triazoles | Sterol biosynthesis in membranes G1: C14-demethylase in sterol biosynthesis (erg11/cyp51) | 250 | Plates assay |
| Beltanol (Su) | Probelte, S.A.U., Murcia, Spain Gadot Agro (Kidron, Israel) | Sulphate 8-Hydroxyquinoline | Sulphuric acid | | | 500 | Plates assay |

**Table 1.** *Cont.*

| Fungicide Name and Acronym | Manufacturer Supplier | Active Ingredient (Common Name) | Group Name | Chemical Group | Target Site of Action | AI (g/l) | Test [2] |
|---|---|---|---|---|---|---|---|
| Octave (Pr-mc) | BASF (Ludwigshafen, Germany)<br><br>Merhav Agro Ltd. (Herzliah, Israel) | Prochloraz present as the manganese chloride complex | DMI-fungicides (demethylation inhibitors) | Imidazoles | C14-demethylation in sterol biosynthesis (erg11/cyp51) | 500 | Plates assay |
| Terraclor super X (Pi-Et) | Amvac (Los Angeles, CA, USA)<br><br>Luxembourg Industries Ltd. (Tel Aviv, Israel) | Pintachloronitrobenzene (PCNB) | AH-fungicides (aromatic hydrocarbons) (chlorophenyls, nitroanilines) | Aromatic hydrocarbons | Cell peroxidation (proposed) | 232 | Plates assay |
| | | Etridiazole | Heteroaromatics | 1,2,4-thiadiazoles | Cell peroxidation (proposed) | 58 | |
| Ohayo (Fluaz) | Phyteurop (Montreuil-Bellay, France)<br><br>Luxembourg Industries Ltd. (Tel Aviv, Israel) | Fluazinam | QiI-Quinone inside inhibitors | 2,6-dinitro-anilines | Respiration C5: uncouplers of oxidative phosphorylation | 500 | Plates assay |
| Amistar (Az) | Syngenta (Basel, Switzerland)<br><br>AdamaMakhteshim (Airport City, Israel) | Azoxystrobin | QoI-fungicides (quinone outside inhibitors) | Methoxy-acrylates | Respiration C3:cytochrome bc1(ubiquinol oxidase) at Qo site (cyt b gene) | 250 | Plates assay |
| Sportak (Pr) | Merhav Agro Ltd. (Herzliah Israel)<br><br>Makhteshim Agan (Airport City, Israel) | Prochloraz | DMI-fungicides (demethylation inhibitors) | Imidazoles | C14-demethylation in sterol biosynthesis (erg11/cyp51) | 450 | Full Season plants |

**Table 1.** *Cont.*

| Fungicide Name and Acronym | Manufacturer Supplier | Active Ingredient (Common Name) | Group Name | Chemical Group | Target Site of Action | AI (g/l) | Test [2] |
|---|---|---|---|---|---|---|---|
| Azimut (Az-Te) | Adama Makhteshim (Be'er Sheva, Israel) | Azoxystrobin 12% | QoI-fungicides (quinone outside inhibitors) | Methoxy-acrylates | Respiration C3: cytochrome bc1 (ubiquinol oxidase) at Qo site (cyt b gene) | 120 | Full Season plants |
| | | Tebuconazole 20% | DMI-fungicides (DeMethylation Inhibitors) (SBI: Class I) | Triazoles | C14-demethylase in sterol biosynthesis (erg11/cyp51) | 200 | |
| Vibrance (Fl-Se) | Syngenta (Basel, Switzerland) Gadot Agro (Kidron, Israel) | Fludioxonil 2.5% | PP-fungicides (PhenylPyrroles) | Phenylpyrroles | MAP/HistidineKinase in osmotic signal transduction (os-2, HOG1) | 25 | Full Season plants |
| | | Sedaxen 2.5% | SDHI (succinatedehydrogenase inhibitors) | pyrazole-4-carboxamides | Complex II: succinate-dehydrogenase | 25 | |

[1] The information is based on the manufacturers' publications and the report of the Fungicide Resistance Action Committee (FRAC 2022). [2] The fungicides tested in plates assay were evaluated, for the first time in Israel, against the onion basal rot disease (FBR) pathogens, *F. oxysporum* f. sp. *cepae* (B14 strain) and *F. acutatum* (B5 strain). Three pesticides (Pr, Az-Te, and Fl-Se) were previously tested in plates, seeds, and sprouts assays and evaluated here in a full growing season.

### 2.2. Examination of Selected Pesticides in Pots under Field Conditions throughout a Full Growing Season

#### 2.2.1. Growth Protocol and Conditions

To examine the effect of chemical FBR control in pots, we used substances that were previously [15] found to be effective in the plates inhibition test: Prochloraz (Pr), Azoxystrobin + Tebuconazole (Az-Te), and Fludioxonil + Sedaxen (Fl-Se) (Table 1). An effective compound had a high statistical difference from the control ($p \ll 0.05$), even at the lowest dosage tested. The experiment was carried out at the Avnei Eitan Experimental Farm in the Golan Heights (north-eastern Israel, $32°49'03.3''$ N $35°45'46.4''$ E) in 10-L pots placed in an open area. The pots were watered with computerized drip irrigation at the recommended dosages according to the growing protocol (in spring and summer, 1 L/pot daily). Fertilization and insecticide treatments were carried out following the recommendations of the Ministry of Agriculture. The pots were filled with infected field soil from two agricultural fields in the Golan Heights: soil from a red onion growing field, Noam variety (Moshav Eliad plot), and soil from a white onion growing field, Orlando variety (Kibbutz Ortal plot). Perlite number 4 (at a ratio of 1:3) was added to this ground for aeration. In each pot, five onion seeds were buried 2 cm deep in the soil (in the soil of a red onion field—Noam, and in the soil of a white onion field—Orlando). The experimental and control groups comprised 10 repetitions (pots), so a total of 260 pots were used.

#### 2.2.2. Important Dates and Meteorological Data

The chemical control semi-field experiment was conducted in the winter-summer of 2022. The key dates are detailed in Table 2. The temperatures and humidity parameters during the onion growing season varied significantly between the winter (post-sprouting period) and the spring-summer (growing period). The meteorological data are summarized in Table 3.

**Table 2.** The experiment's dates.

| Date | Inoculation, Planting, and Sprouting Assessment | Days from Sowing |
|---|---|---|
| 12 January 2022 | 1st inoculation (sterilized infected wheat grains) | −8 |
| 20 January 2022 | Sowing and 2nd inoculation (2 discs per seed) | 0 |
| 10 February 2022 | 3rd inoculation (2 discs per sprout) | 21 |
| 20 March 2022 | Aboveground sprouting | 115 |
| | | **Days from sprouting** [1] |
| 23 March 2022 | Emergence evaluation I | 3 |
| 3 April 2022 | Emergence evaluation II | 14 |
| 24 April 2022 | Emergence evaluation III | 35 |
| **Pesticide treatments** | | |
| 5 April 2022 | Pesticide I | 16 |
| 24 April 2022 | Pesticide II (20 days from Pesticide I) | 35 |
| 15 May 2022 | Pesticide III (20 days from Pesticide II) | 56 |
| 10 May 2022 | Bifenthrin (Talstar) spraying treatment [2] | 51 |
| **Sampling and harvest** | | |
| 24 May 2022 | Mid-season sampling and thinning | 65 |
| 13 July 2022 | Harvest and final sampling | 115 |

[1] The aboveground sprouting was delayed due to the low temperatures during the winter. Therefore, 20 March 2022 was designated as the sprouting (above soil surface peek) day. [2] Treatment against onion fly maggots (*Delia antiqua*). The preparation is marketed by Luxembourg Industries Ltd., Tel Aviv, Israel.

**Table 3.** Meteorological data [1].

| Parameters | Winter (Post-Sprouting Period) | Spring-Summer (Growing Period) |
|---|---|---|
| Dates | 20 January–19 March | 20 March–13 July |
| Temperature (°C) | 9.5 ± 3.8 | 21.2 ± 7.2 |
| Humidity (%) | 74.0 ± 16.9 | 51.0 ± 24.8 |
| Precipitation (sum mm) | 100.1 | 42.7 |

[1] Data (average ± standard deviation or sum) from the Avnei Eitan Meteorological Station, Soil Conservation Division, Israeli Ministry of Agriculture.

### 2.2.3. Complementary Inoculation

The inoculation method consisted of naturally infected soil and complementary infection conducted in three steps to guarantee a high and uniform infection load. First, pre-sowing soil was inoculated with sterilized infected wheat grains that had been prepared as previously described [20,21]. Briefly, the wheat seeds were soaked in tap water overnight, filtered, and dried. The wheat seeds were sterilized by autoclave for 30 min at 120 °C, and 150 g was incubated with 10 *Fusarium* sp. mycelium agar disks (see Section 2.1) in sterilized 0.5 L plastic boxes for 2–3 weeks at 28 ± 1 °C in the dark. Separate infected grains batches were prepared for each species (with 10 colony agar disks from a single culture). We also set up a mixed *Fusarium*-infected wheat grains batch (with 5 colony agar disks from each of the two species cultures).

The inoculated wheat seeds were mixed vigorously by shaking every two days to ensure equal seed infection. The soil was inoculated by mixing 20 g of the sterilized infected wheat grains with the soil top 10 cm. Second, two mycelia discs (see Section 2.1) from *F. oxysporum* f. sp. *cepae,* or *F. acutatum,* or a combination of both pathogens (one disk from each fungus) were added to each plant during and after the sowing (Table 2).

### 2.2.4. Pesticide Treatments

The pesticides were applied to the pots according to timeline in Table 2. The control was infected pots without pesticides. The pots that received Prochloraz or Az-Te treatments were divided into two subgroups of five pots—one received high-dose pesticide treatments (0.3%) and the second a low dose (0.15%). The pesticides were applied to the 1 L/pot irrigation water at three intervals (16, 35, and 56 days from sowing) in the above concentrations. The preparation Fl-Se was applied by seed coating: 1 μL preparation dissolved in 6.5 μL water for 1 gr seeds (ca. 300 seeds), i.e., 0.003 microliters per seed. According to the manufacturer's recommendation, the seeds were mixed vigorously with the fungicide solution until completely covered.

### 2.2.5. Growth and Disease Estimation

Above-soil-surface emergence was evaluated after 3, 14, and 35 days from the day of sprouting (Table 2). The sowing was on 20/1/22, but the aboveground germination was delayed due to the cold winter. Therefore, 20 March 2022, was designated as the emergence day. After another 65 days and at the end of the growing season (day 115), growth indices were measured, and the disease symptoms (mortality and dehydration) were estimated. The collected growth parameters indicate the severity of the disease and the effectiveness of the treatments. They include the number of surviving plants, the weight of the onion bulbs, the number of leaves, shoot height, and shoot fresh weight. Also, the plants' shoot water content was set by subtracting the plants' dry weight from their wet biomass weight.

### 2.3. Statistical Analysis

Statistical analysis was done using the JMP program, 15th Edition (SAS Institute Inc., Cary, NC, USA). The data analysis (of the media plates' colony diameter, growth indexes, and disease symptoms) was done using one-way analysis of variance (ANOVA) with a post-hoc comparison based on a *t*-test and a significance level of $p < 0.05$.

## 3. Results

### 3.1. Fungicides Culture Plates Assay

The effect of selected pesticides in inhibiting the onion *Fusarium* basal rot (FBR) pathogens, *F. oxysporum* f. sp. *cepae* (B14) and *F. acutatum* (B5), in culture plates was measured after two-day incubation. This scan showed that the fungicides Azoxystrobin + Difenoconazole (Az-Di) mixture, Fluopyram + Trifloxystrobin (Fluop-Tr) mixture, Fluazinam (Fluaz), and Difenoconazole (Di) have significant pest control potential ($p << 0.05$) in inhibiting both pathogens (Figure 1). These preparations caused significant development inhibition even at the lowest concentration (1 ppm). Most of the other substances tested caused a similar effect at concentrations of 100 ppm.

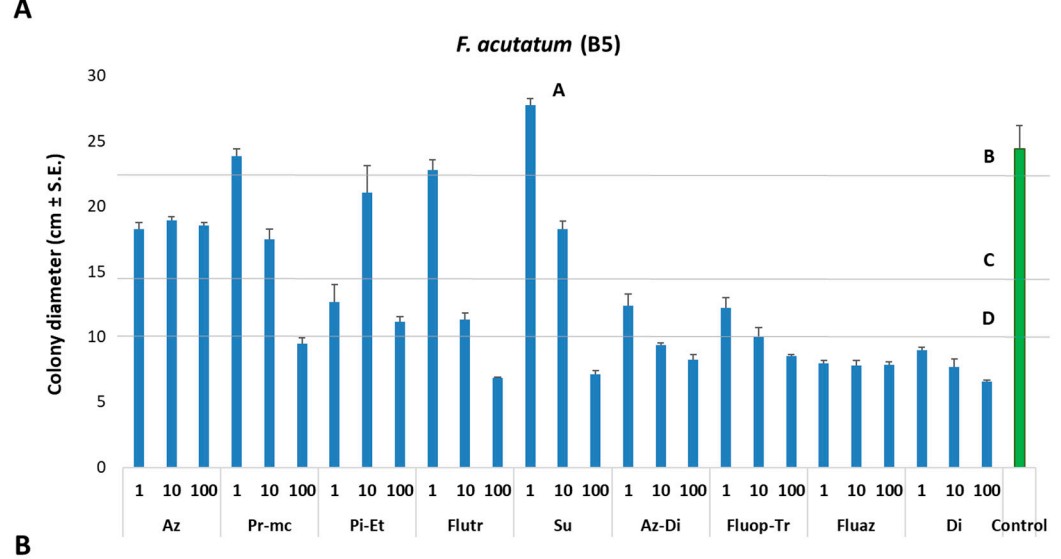

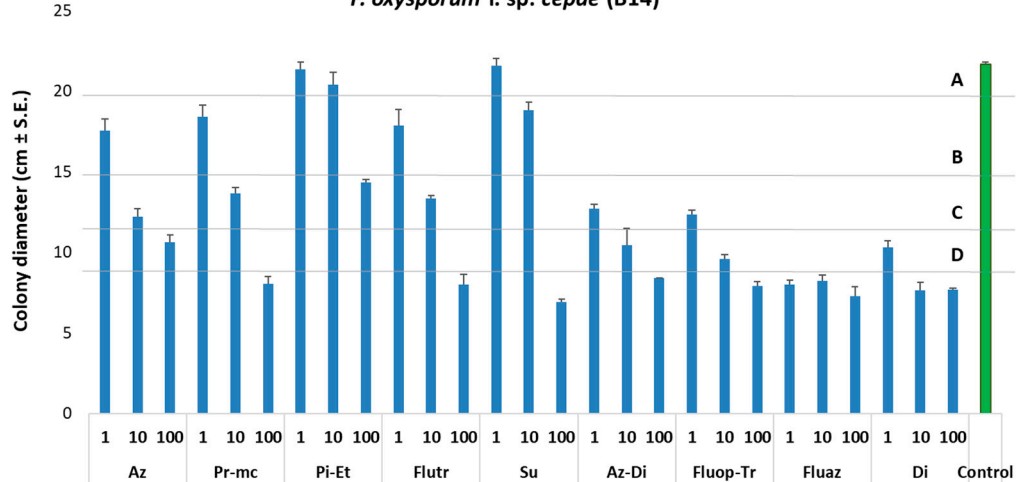

**Figure 1.** The effect of selected pesticides on the development of *F. oxysporum* f. sp. *cepae* (**A**, strain B14) and *F. acutatum* (**B**, strain B5) in culture plates. The pesticides were tested at concentrations of 1, 10, and 100 ppm. Mycelial discs from a 8-day-old colony were sown on potato dextrose agar (PDA) medium plates that contained pesticides in three concentrations and incubated for two days. Controls (green columns) are PDA without fungicides. Bars represent the mean colony diameter of four replicates, and error bars signify a standard error. Different letters (A–D, on the chart's right side) represent a significant difference ($p < 0.05$) in the ANOVA test.

### 3.2. Examination of Selected Pesticides in Pots under Field Conditions throughout a Full Growing Season

3.2.1. Impact of the Treatments on the Plants' Soil Surface Germination and Survival

Three pesticides were selected for the pot trial under field conditions: Prochloraz (Pr), Azoxystrobin + Tebuconazole (Az-Te), and Fludioxonil + Sedaxen (Fl-Se) (Table 1). These preparations were tested in a previous study [15] in seeds (in vitro) and sprouts in pots in a controlled growing room. The pots in the Prochloraz and Az-Te treatments were divided into two subgroups of five pots—one received pesticide treatments at a high dose (0.3%) and the other at a low dose (0.15%) applied 16, 35, and 56 days after sowing. The Fl-Se preparation was applied by seed dressing (see Section 2.2.4). Soil surface germination occurred about two months after sowing when the temperatures warmed up (Figure 2).

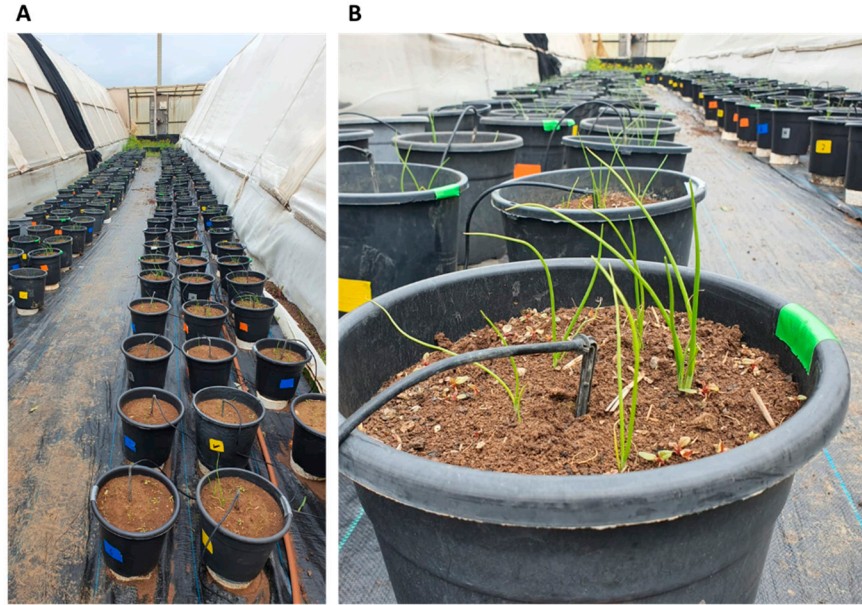

**Figure 2.** The semi-field pot experiment setting (**A**) and the 3rd-day aboveground sprouting (**B**). Two onion varieties were tested in a pot trial under field conditions at the Avnei Eitan Experimental Farm in the Golan Heights (north-eastern Israel): white Orlando cv. (**A**, left side pots) and red Noam cv. (**A**, right side pots). The pots were infected separately with each pathogen, *F. oxysporum* f. sp. *cepae* (B14) and *F. acutatum* (B5), and tested against a combination of the two pathogens and an unprotected control. Three pesticides were inspected as protective treatments: Prochloraz (Pr), Azoxystrobin + Tebuconazole (Az-Te), and Fludioxonil + Sedaxen (Fl-Se) (Table 1). The pots in the Prochloraz and Az-Te treatments were divided into two subgroups of five pots—a high-dose (0.3%) and a low-dose (0.15%) pesticide treatment applied 16, 35, and 56 days after sowing. The Fl-Se preparation was applied by seed dressing (0.003 microliters per seed).

Three seedling survival and emergence evaluations were made, 3, 14, and 35, after the beginning of the soil surface appearance (Figure 3). The first chemical treatment was performed 16 days after the soil surface peeks started, so the first two emergence evaluations were performed before this treatment, and the third was done afterward. Still, in each onion cultivar, the three soil surface peek evaluations (done in the same way, without referring to the high and low pesticide concentrations applied before the third measurement) gave similar results. The only difference was an overall decrease in the sprouts' numbers and more identifiable statistical differences between the treatments in the third evaluation. Does this mortality increase a consequence of the chemical first treatment's phytotoxicity? Apparently not. The general lower survival parameters after 35 days were also measured in the chemical-free control and were evidenced between the two first evaluations made before the chemicals were added. For example, the *F. acutatum* (B5) control soil surface peek

values were decreased from ca. 60% (day 3) to 40% (day 14) and 30% (day 35). Similarly, the *F. oxysporum* f. sp. *cepae* (B14) control values dropped from 75% to 60% and 40%.

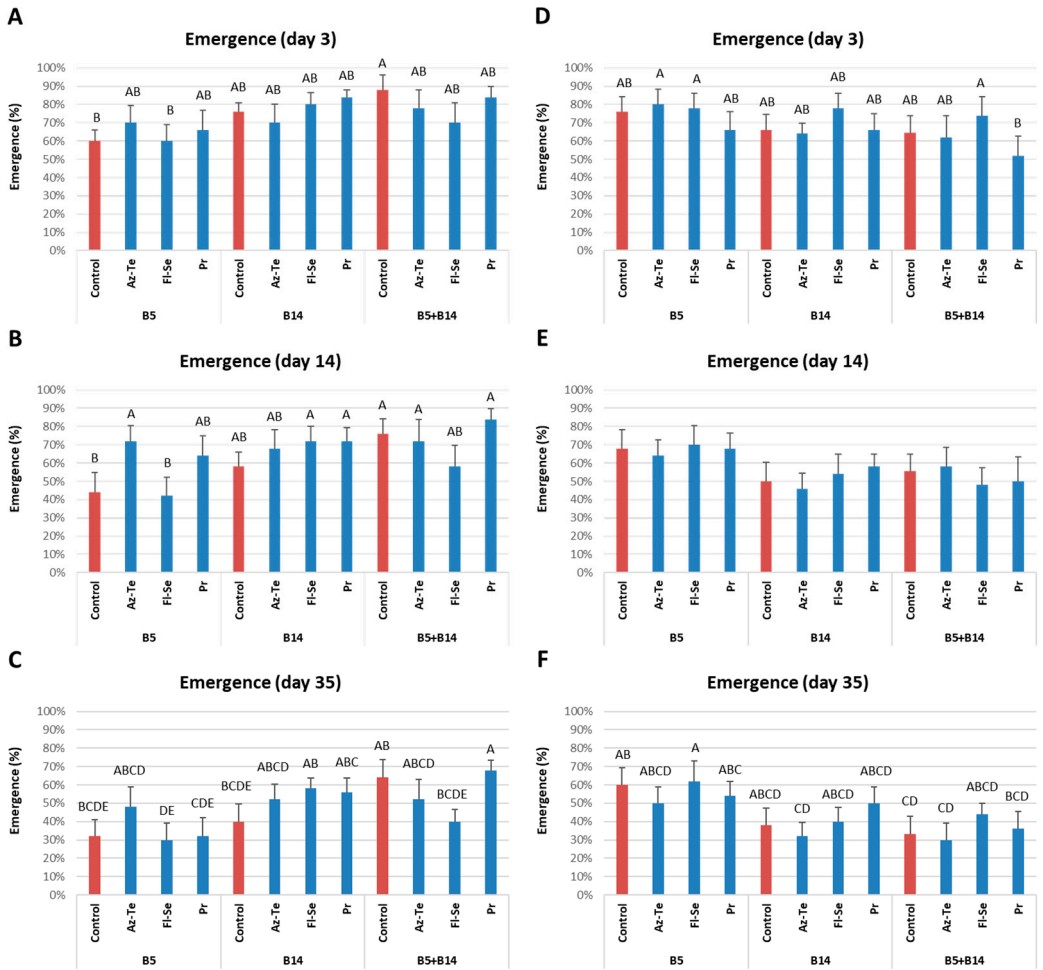

**Figure 3.** The effect of chemo-pesticide treatments on the emergence and survival of white onion plants (Orlando cv., **A–C**) and red onion plants (Noam cv., **D–F**) 4, 14, and 35 days after the beginning of the soil surface appearance. B5-*F. acutatum*, B14-*F. oxysporum* f. sp. *cepae*. The experiment and abbreviations are described in Figure 2. Bars indicate the mean of 10 biological replications (pots). Controls (red columns) are plants without fungicides. Error bars are standard errors. If it exists, different letters (A–E, above the chart's bars) represent a significant difference (*p* < 0.05) in the ANOVA test.

Thirty-five days after the first soil surface germination, aboveground peek values (Figure 3) were similar to the findings in a growth chamber in a previous study on day 30 [15]. The Prochloraz preparation prevented the inhibition of soil surface germination caused by *F. oxysporum* f. sp. *cepae* (B14 strain). This effect was found in both onion varieties (white Orlando cv., Figure 3A, and red Noam cv., Figure 3B) but was not statistically significant compared to the unprotected control. Some treatments inhibited the plants' development (not significantly), for example, Fl-Se in the white Orlando cv. onion and Az-Te in the red Noam cv. onion.

### 3.2.2. Impact of the Treatments at the Mid-Season Sampling (Day 65)

Growth estimation made at the end of the sprouting phase reveals the influence of the anti-fungal treatments. Prochloraz at the low concentration (0.15%) was the best treatment (with a significant difference, *p* < 0.05, in some parameters) compared to the control in the *F. oxysporum* f. sp. *cepae* (B14)-infected plants regarding the plants' fresh weight, shoot

length and phenological development (number of leaves, Figure 4). This chemo-shielding was effective in both onion cultivars tested (Figures 4 and 5). The lesser effectiveness of the Prochloraz high concentration (0.3%) is most likely a result of overdosage, which may cause phytotoxicity. In fact, this was the case in the Az-Te treatment that failed to protect the plants, especially at the higher concentration applied. The Fl-Se treatment may have some protective ability in the red onion plants (Noam cv., Figure 5) against the pathogens tested. Such potential is not reflected in statistical significance compared to the control. Still, the treatments' full potential will be revealed at the season's end.

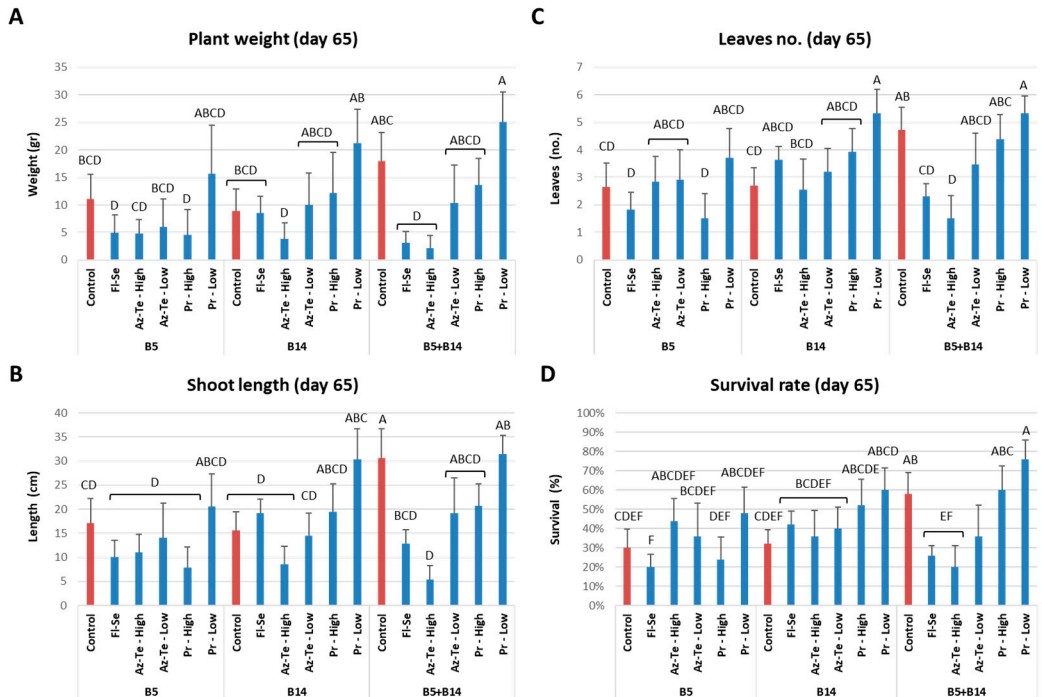

**Figure 4.** Effect of pesticide treatments on the growth indexes and survival of white onion plants (Orlando cv.) 65 days after soil surface germination. The experiment and abbreviations are described in Figure 2. The mid-season evaluation included the plants' weight (**A**), shoot length (**B**), and phenological development (number of leaves, **C**). Each value is an average of four plants/pot. The survival rate (**D**) is the percentage of five plants/pot. B5-*F. acutatum*, B14-*F. oxysporum* f. sp. *cepae*. Bars indicate the mean of five (Prochloraz and Az-Te) or 10 (Fl-Se) biological replications (pots). Controls (red columns) are plants without fungicides. Error bars are standard errors. Different letters above the chart's bars (A–F) represent a significant difference ($p < 0.05$) in the ANOVA test.

A most interesting understanding was achieved by analyzing only the controls' results (without chemical intervention) in both the soil surface germination (day 35, Figure 3) and mid-season (day 65, Figures 4 and 5) data collection stages. It appears that *F. oxysporum* f. sp. *cepae* (B14) is more aggressive towards the red onion Noam cv. compared to the white Orlando cv. In contrast, *F. acutatum* (B5) was more virulent to Orlando cv. Furthermore, inoculating the plant with both pathogens led to a different outcome in the two onion varieties. In the white Orlando cv., the double infection resulted in reduced growth suppression, while in the red Noam cv., it caused a severe disease similar to the *F. oxysporum* f. sp. *cepae* sole infection.

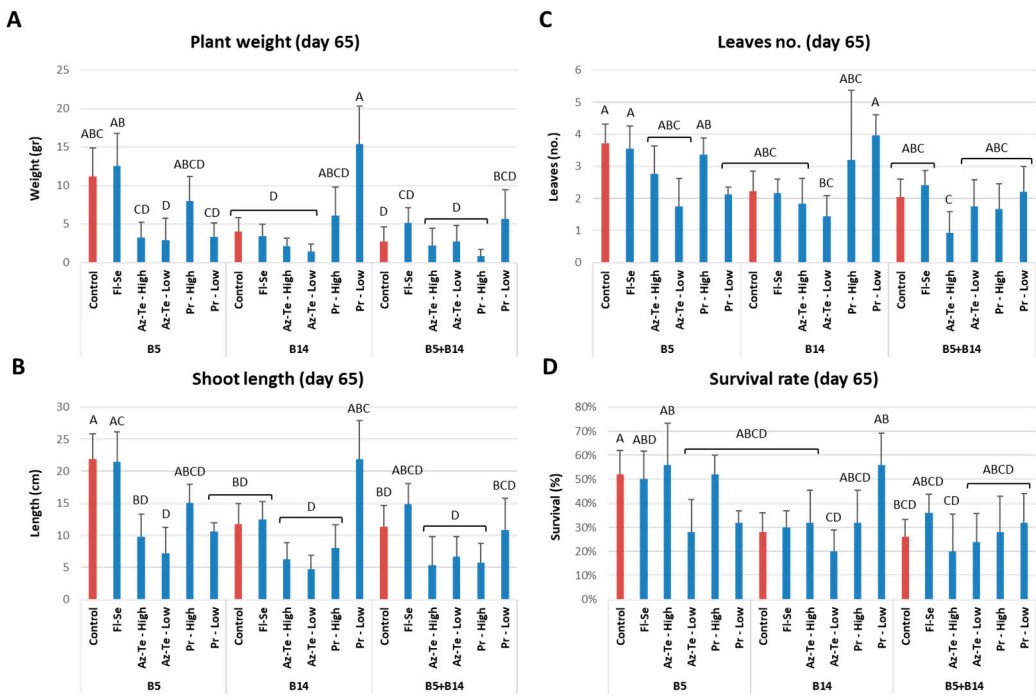

**Figure 5.** Effect of pesticide treatments on the growth indexes and survival of red onion plants (Noam cv.) 65 days after soil surface germination. The experiment and abbreviations are described in Figure 2. The mid-season evaluation included the plants' weight (**A**), shoot length (**B**) and phenological development (number of leaves, **C**). Each value is an average of four plants/pot. The survival rate (**D**) is the percentage of five plants/pot. B5-*F. acutatum*, B14-*F. oxysporum* f. sp. *cepae*. Bars indicate the mean of five (Prochloraz and Az-Te) or 10 (Fl-Se) biological replications (pots). Controls (red columns) are plants without fungicides. Error bars are standard errors. Different letters above the chart's bars (A–D) represent a significant difference ($p < 0.05$) in the ANOVA test.

### 3.2.3. Impact of the Treatments at the Season End (Day 115)

At the end of the season (day 115 after the soil surface germination), many plants of both onion species showed progressive disease symptoms (Figure 6). In severe cases, the disease manifested itself in acute dehydration and death. In milder cases, there was a delay in the development of the aboveground onion parts, accompanied by yellowing and drying of the leaves. The roots showed varying degrees of rotting (in severe cases, there were no roots at all), and the rotting deteriorated and spread to the onion-basal plate and the parts next to it.

The estimation of the two onion varieties' growth parameters under the *Fusarium* spp. infection stress and the protective chemical treatments is presented in Figures 7 and 8. In most parameters, the Prochloraz chemical treatments achieved the highest score in both cultivars. Still, statistical significance compared to the unprotected control wasn't reached. At the harvest, the Fl-Se treatment excelled in protecting the red Noam cv., particularly against *F. oxysporum* f. sp. *cepae* (B14). This treatment significantly improved the plants' shoot wet weight and number of leaves compared to the control ($p < 0.05$, Figure 8). At high doses, the Prochloraz and Az-Te pesticides damaged the development of the plants and crops. The Az-Te treatment, even at the lowest dosage, appears phytotoxic. Thus, it will be necessary to reduce the dose of the preparation to evaluate its protective effect better.

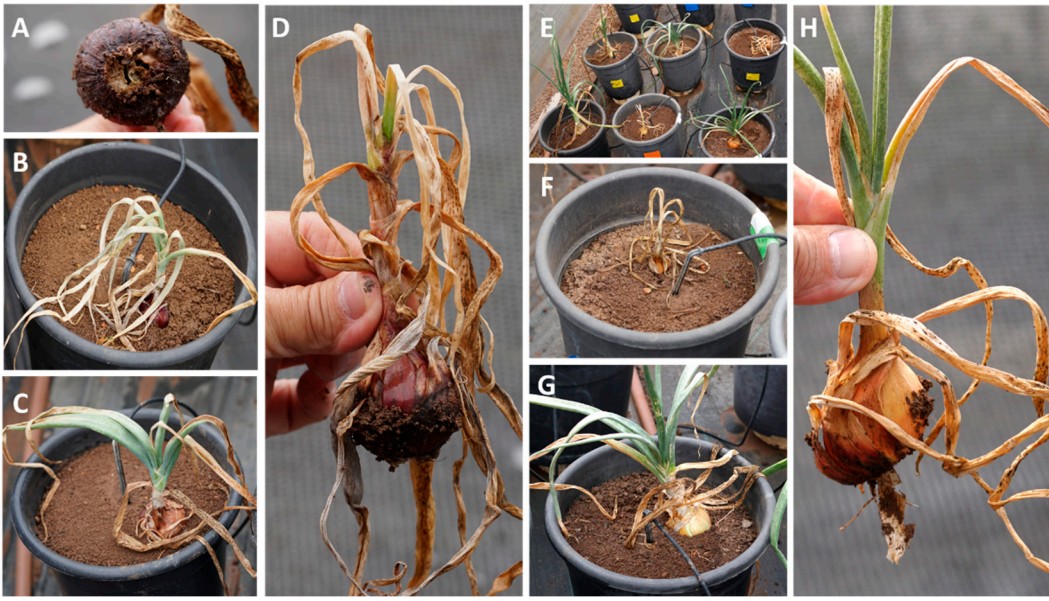

**Figure 6.** The experimental photos at the end of the season (day 115 from soil surface germination). The experiment is described in Figure 2. (**A–D**) onion *Fusarium* basal rot (FBR) disease symptoms in the red Noam cv. plants. (**E–G**) symptoms in the white Orlando cv. plants. (**E**) representative photo of the disease diversity among the different treatments. (**A,B,D,F,H**) severe disease, (**C,G**) mild disease.

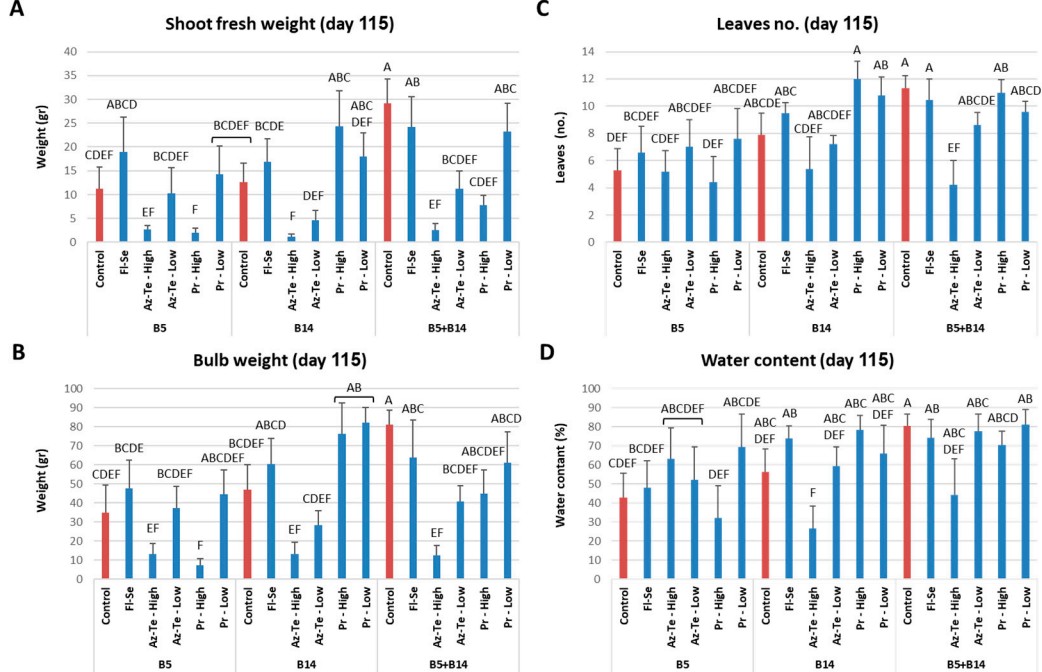

**Figure 7.** Effect of pesticide treatments on onion weight of white onion plants (Orlando cv.) 115 days after germination. The experiment and abbreviations are described in Figure 2. The harvest evaluation included the plants' shoot fresh weight (**A**), bulbs' wet weight (**B**), phenological development (number of leaves, **C**), and water content (**D**). B5-*F. acutatum*, B14-*F. oxysporum* f. sp. *cepae*. Bars indicate the mean of five (Prochloraz and Az-Te) or 10 (Fl-Se) biological replications (pots). Controls (red columns) are plants without fungicides. Error bars are standard errors. Different letters above the chart's bars (A–F) represent a significant difference ($p < 0.05$) in the ANOVA test.

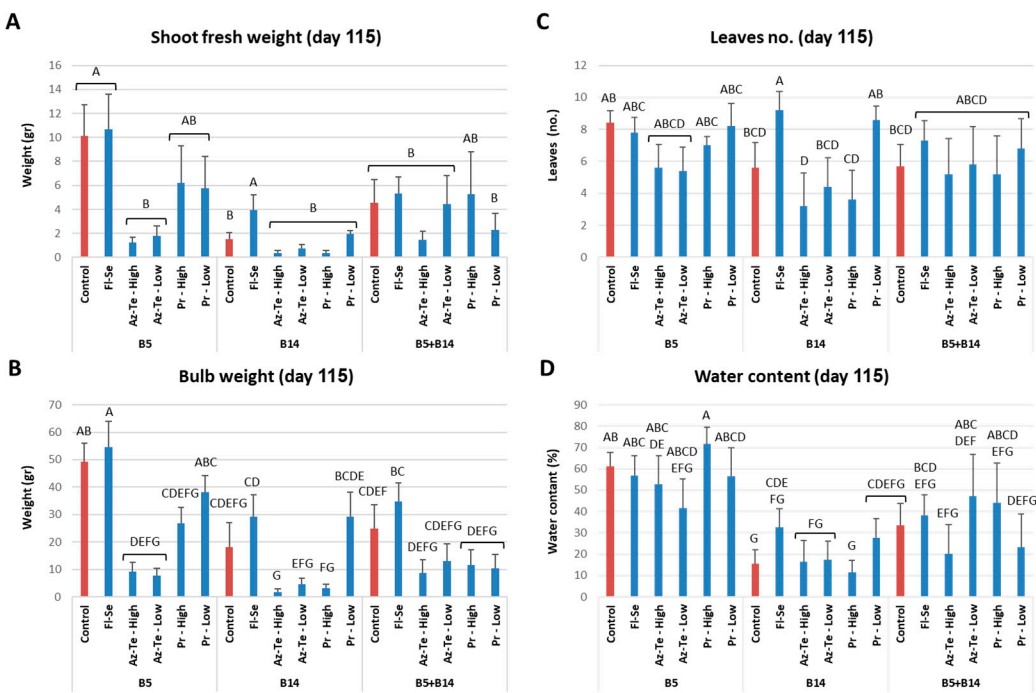

**Figure 8.** Effect of pesticide treatments on shoot weight of red onion plants (Noam cv.) 115 days after germination. The experiment and abbreviations are described in Figure 2. The harvest evaluation included the plants' shoot fresh weight (**A**), bulbs' wet weight (**B**), phenological development (number of leaves, **C**), and water content (**D**). B5-*F. acutatum*, B14-*F. oxysporum* f. sp. *cepae*. Bars indicate the mean of five (Prochloraz and Az-Te) or 10 (Fl-Se) biological replications (pots). Controls (red columns) are plants without fungicides. Error bars are standard errors. Different letters above the chart's bars (A–G) represent a significant difference (*p* < 0.05) in the ANOVA test.

Similar to the mid-season results, the combined infection with the two pathogens caused less damage to the Orlando cv. plants than when inoculated with each separately. This result suggests a mutually antagonistic relationship or competition between *F. oxysporum* f. sp. *cepae* and *F. acutatum*. This trend was less pronounced in the Noam cv.

## 4. Discussion

Beneficial plants cultivated for food, medicinal ingredients, or other purposes suffer from many diseases that result in significant losses to the growers [22]. Despite worldwide scientific efforts to reduce the use of chemical pesticides and adopt eco-friendly control methods [23], traditional fungicides are still our most effective and feasible strategy to protect field crops [24]. Chemical protection is critical in severe diseases where biological control fails to provide a sufficient solution. A new approach that combines the benefits of both chemical and biological methods (integrated pest control) may provide high efficiency with minimal chemical intervention while decreasing the evolution of fungal fungicides resistance [25]. Yet, this method relies on friendly bio-control agents such as *Trichoderma* species that can cope with the chemical ingredients [26].

Moreover, developing such control solutions requires significant efforts and adjustments since they are often tailored to specific diseases and crops. They may not protect other crops in the rotation or affect other phytopathogens. Agrochemical pesticides (especially broad-spectrum preparations) can provide a rapid and more general solution, particularly in situations of new emerging pathogens or the worsening of existing plant diseases [24]. Thus, chemical food control is crucial to support the need of the world's growing population. As done here, the research effort towards that goal should focus on: (1) developing potential preparation up to a mature applicative method (with attention to the dosages, the implementation time schedule, and the integration with other treatments);

and (2) continuing to seek and test (fast screening) new compounds. This is especially important since fungicides' resistance is becoming increasingly serious [27].

Already there are confirmed reports on resistance in the *Allium* specialized *Fusarium* population [8] and resistance to many fungicides (i.e., Carbendazole, Benomyl, Thiophanate-methyl, Thiabendazole, Fludioxonil, and Prochloraz) is reported in pathogenic *Fusarium* populations in other plants [8]. Actually, Prochloraz, a leading treatment for onion *Fusarium* basal rot (FBR) found in this work, can evoke responsive genes enabling DMI resistance that have already been described in other phytopathogens (see, for example, [28,29]).

Today, the coping means applied in Israel against FBR disease are few and include a four-year growing cycle and disinfecting the soil with metam sodium [2]. At the same time, agricultural tools (harrows, plows, etc.), contaminated equipment, and the workforce unintentionally allow the disease to continue and spread to new growing areas [8]. The current study examined the potential of chemical control to reduce the damage of this disease. First, new effective substances were found against the pathogens involved. Second, selected preparations from those previously tested in seedlings [15] were tested here in a full growing season. One of them, Prochloraz, showed effectiveness against one of the main causal agents of the onion FBR disease, the pathogen *F. oxysporum* f. sp. *cepae*. At the same time, it was less effective against the pathogen *F. acutatum*.

Prochloraz was dominant as a successful treatment in controlling *F. oxysporum* f. sp. *cepae* (B14) was well documented during the growth period up to 65 days from the surface peeking, in both onion cultivars tested. Yet at the season end (day 115), the treatment's impact was more pronounced in the bulbs' fresh weight and less in protecting the shoot. Most plants' upper parts at this growth stage suffer from dehydration (shoot wet biomass reduction), and exclusively the Fl-Se treatment could protect them. The difference between the two fungicides may be their residuals in plant tissues after the third application (day 56). Therefore, a preferred choice may be alternating Prochloraz with Fl-Se towards the season-ending. Such a solution should be tested in future works.

Interestingly, similar results were achieved after 30 days in sprouts in a growth room trial [15] and in open-air enclosure pots, carried out here. In a previous work done in seedlings, the Prochloraz treatment in Noam cv. reduced the harmful impact of *F. oxysporum* f. sp. *cepae*. on the plants' wet biomass. Still, it was ineffective in the Noam cv. against *F. acutatum*. Here, too, Prochloraz was the most efficient, but it failed to protect the Noam cv. plants against *F. acutatum*. These findings imply a good predictive ability for the growth room trial in assessing new potential FBR chemo-pesticides. It is an important conclusion since conducting outdoor experiments over a full season requires significant investment and time.

Regarding the Fludioxonil + Sedaxen mixture (Fl-Se) pesticide, it seems that the preparation has a wide range of effectiveness against the two *Fusarium* species. This fungicide achieved partial (significant) success in the Noam cv. plants infected with *F. oxysporum* f. sp. *cepae*. There is a good possibility that the Fl-Se treatments' results are the consequence of the concentration or application method we used in a seed dressing, especially since this onion variety is probably more sensitive than the white Orlando cv. onion to the chemical treatments. So, different concentrations of the preparation and different application methods must be tested to optimize its effect. Also, the potential of this preparation to protect the onion cultivars against various *Fusarium* species, i.e., *F. proliferatum* and *F. anthophilium* (other members in the FBR complex [2]), should be explored further in subsequent studies.

The different growth measures used in the current study showed similar tendencies (for instance, see Figures 7 and 8). This is not surprising because the main FBR symptom at the maturity phase is dehydration of the plant's upper parts (including the leaves and flowers) [8]. As a result of the water content drop, the plants lose weight, their development is disrupted (they are shorter and have fewer leaves), and their bulb development is delayed. Successful chemical protective treatments such as Prochloraz and Fl-Se are reflected in recovering these indices, as demonstrated here.

The Azoxystrobin + Tebuconazole mixture (Az-Te) treatment was probably phytotoxic and damaged the plants' development even at the lowest dosage. The phytotoxic effect of anti-fungal compounds on the cultivated plants varied depending on the dose, the application method and timing, and the test cultivar. So, a delicate adjustment should be tailored in each chemo-control protocol to achieve maximum efficiency and minimum plant toxicity. Such a task becomes more complex and challenging when a few fungal pathogens species involved with each have a different response to the pesticide (as demonstrated here).

*Fusarium* basal rot is associated with different complexes of several *Fusarium* species, of which *Fusarium oxysporum* f. sp. *cepae* and *F. proliferatum* are the most prevalent world-wide [8]. Indeed, one of the challenges in this disease management is that these pathogenic complexes vary in virulence and composition depending on the growing area and host resistance. Intriguing was the discovery that *F. oxysporum* f. sp. *cepae* is more virulent to red Noam cv. and *F. acutatum* (B5) is more aggressive towards Orlando cv. This finding reflects well the origin of these pathogens. *F. oxysporum* f. sp. *cepae* was previously isolated by us from red onion variety (565/505 cv.) infected bulbs, while *F. acutatum* originated from the white Orlando cv. [2]. It seems logical that the co-inoculation of the Noam cv. plants with both pathogens will result in severe disease as the sole *F. oxysporum* f. sp. *cepae* infection (or more), but why did the double inoculation lead to the opposite trend in the Orlando cv.? Antagonistic interactions could result as a consequence of competition for resources such as food and growing space. There are many examples in the literature of such pathogenic cross-talk.

One example is the antagonistic interactions between *Magnaporthiopsis maydis* and *Macrophomina phaseolina*, the charcoal rot disease agent in cotton plants [30,31]. Moreover, *M. maydis* is similarly associated with *Fusarium oxysporum*. But the results of such interactions are not always inhibition since *M. maydis,* together with *Fusarium verticillioides* and *M. phaseolina* can cause devastating maize post-flowering stalk rot diseases [32]. Another example is the cross-talk between *F. oxysporum* and *Fusarium solani* on pea roots [33] in which these phytopathogens are antagonists to one another. In contrast, roots infected with *Aphanomyces euteiches* were more susceptible to *Fusarium* root rot than those exposed only to *Fusarium* spp. [34].

Thus, a pest control plan must refer to the plant variety and its degree of resistance, the growing conditions, and the microflora that distinguishes it and is involved in causing the disease. In the case of onion FBR disease, in which several violent *Fusarium* species are involved [8], the relationship between them and the possible involvement of other *Fusarium* species (not yet identified) are crucial. An accurate valuation of the importance of each species in the *Fusarium* disease complex is challenging. Even more, pathogens from other species, such as *Colletotrichum* spp., *Alternaria* spp., *Peronospora* spp., *Botrytis* spp. and *Phoma* spp. [35], must be taken into account. Since the onion varieties show variable resistance both to the toxicity of the pesticides and to the different *Fusarium* species involved in the disease, we must refer to these aspects when planning a protective suit adapted to each variety. It is, of course, essential to pay attention to the time and method of application of the antifungals and to try to combine pesticides with different mechanisms of action to prevent the development of resistance against the preparation.

Continuing the effort to locate and apply pesticides is essential. It may improve our ability to deal with a disease that poses a growing risk to the onion industry in the short and long term. It is necessary to continue testing pesticides while evaluating their toxicity to the plant and their effectiveness against each of the pathogens involved in causing the disease. It must be remembered that the early tests in substrate plates that aim at quickly scanning many pesticides and pointing out the most effective ones have a limited predictive ability to field situations [14]. The most promising compounds should be tested later on in seeds and sprouts while eliminate ineffective preparations [36]. Only in the final stage will a test be carried out throughout a full growing season in pots under field conditions, with the aim of testing the effectiveness of pesticides that have successfully passed the preliminary

tests. This method is essential in order to reduce the large investment and long duration of time involved in experiments carried out in field conditions throughout an entire growing season.

Effective control methods can be adopted from other countries but must be evaluated and validated against the local *Fusarium* population in Israel. For instance, in Finland, the incidences of *Fusarium* spp. infected bulbs are up to 20% despite chemical treatments [37]. This is probably due to a significant variation in *Fusarium* populations and environmental factors. Thus, there is no assurance that the same method will have similar effectiveness anywhere. Field evaluation must address not only the short-term FBR response, but also the influence on the soils' and plants' microflora communities, including the recolonization of soil [38].

## 5. Conclusions

Onion *Fusarium* basal rot (FBR) results in yield and quality losses, both before and after the harvest, in many onion fields around the globe. Disease incidence differs considerably [39], depending on the virulence of the *Fusarium* species involved, the environmental conditions, the degree of host susceptibility, and the phenological stage. The disease can occur at all stages of onion development [40], but sprouts and mature post-harvest bulbs are the most susceptible [39]. Despite implementing a four-year rotation and soil disinfection, the disease spread in Israel has increased significantly in recent years. The current study aimed at examining the potential of fungicides to restrict the FBR leading causal agents in Israel.

First, new effective formulas were found against the pathogens involved. The fungicides Azoxystrobin + Difenoconazole mixture, Fluopyram + Trifloxystrobin mixture, Fluazinam, and Difenoconazole have significant control potential against *F. oxysporum* f. sp. *cepae* (B14) and *F. acutatum* (B5). Second, selected compounds from those previously tested in seedlings [15] were evaluated in an entire growing season. In a semi-field pot experiment on day 35, Prochloraz preparation prevented the sprouting inhibition by *F. oxysporum* f. sp. *cepae* (B14 strain) in both onion varieties tested (white Orlando cv. and red Noam cv.). This protection continued at the mid-season (day 65) evaluation and on harvest (day 115). At the low concentration (0.15%), Prochloraz rescued the growth parameters of both onion cultivars under *F. oxysporum* f. sp. *cepae* stress. Another preparation, Fludioxonil + Sedaxen (Fl-Se), showed a potential to protect both onion cultivars against the two *Fusarium* species tested. This finding implies that the Fl-Se treatment may have a broad influence, an assumption that should be explored more in follow-up studies.

Interestingly, *F. oxysporum* f. sp. *cepae* was more aggressive towards the red onion Noam cv. while *F. acutatum* was more virulent to the white Orlando cv. This finding reflects well the origin of these pathogens. *F. oxysporum* f. sp. *cepae* was isolated from red onion variety (565/505 cv.), while *F. acutatum* originated from the white Orlando cv. [2]. Furthermore, inoculating the plant with both pathogens led in the red Noam cv. to a severe disease similar to the *F. oxysporum* f. sp. *cepae* sole infection but to reduced disease symptoms in the white Orlando cv. Thus, antagonistic interactions within the *Fusarium* population may exist in some onion genotypes.

The current work advances our understanding of the nature of this plant disease and reveals new ways of controlling it. Future efforts must be dedicated to completing the fungicides' development to a final field protocol. This includes setting the optimum concentration, application method, integration with other control methods, effectiveness against other *Fusarium* species in the FBR complex [2], and compatibility with other treatments in the field.

**Author Contributions:** Conceptualization, O.D., E.D., A.G., S.G. and E.M.; data curation, O.D., E.D. and A.G.; formal analysis, O.D., E.D. and A.G.; funding acquisition, O.D., S.G. and E.M.; investigation, O.D., E.D. and A.G.; methodology, O.D., E.D., A.G. and S.G.; project administration, O.D.; resources, O.D., S.G. and E.M.; supervision, O.D. and S.G.; validation, O.D. and E.D.; visualization, O.D. and E.D.; writing (original draft), O.D.; writing (review and editing), O.D., E.D., A.G., S.G. and E.M. All authors have read and agreed to the published version of the manuscript.

**Funding:** This work was supported by a one-year research grant (2021) from the Israel Organization of Crops and Vegetables, Ministry of Agriculture and Rural Development.

**Institutional Review Board Statement:** Not applicable.

**Informed Consent Statement:** Not applicable.

**Data Availability Statement:** The datasets generated and/or analyzed during the current study are available from the corresponding author on reasonable request.

**Acknowledgments:** We would like to thank Galia Shofman (Migal—Galilee Research Institute and Tel-Hai College, Israel) for her technical assistance, and Oria Reshef, manager of the Avnei Eitan Experimental Farm, for his essential help in performing the farm experiment.

**Conflicts of Interest:** The authors declare no conflict of interest.

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
