# Peer review of "Prevention and Control of Fusarium spp., the Causal Agents of Onion (Allium cepa) Basal Rot"

_horticulturae, doi:10.3390/horticulturae8111071_

Round 1

Reviewer 1 Report

Overall the manuscript explain the comprehensive study on controlling FBR using chemical compounds. Some revisions need to improve the manuscript. Please find the detail review on the attached file and elaborate the comments. 

Author Response

Responses to Reviewer 1’s comments

We thank the reviewer for investing substantial efforts, which undoubtedly contribute to this manuscript. The remarks and suggestions improved this paper’s scientific soundness and accuracy. Your contribution is greatly appreciated.

Overall the manuscript explains the comprehensive study on controlling FBR using chemical compounds. Some revisions need to improve the manuscript. Please find the detailed review in the attached file and elaborate on the comments.

Reply: Thank you for the positive evaluation of our manuscript. All your remarks and suggestions were addressed carefully and thoroughly, as detailed below.

Line 94 - a (not capital letter). Please check others.

Reply: Corrected to a small letter as advised. Thank you. We carefully checked the entire manuscript for similar capital letter mistakes and corrected them.

Line 119 - a (small letter).

Reply: Corrected to a small letter as advised.

Line 119 - Please explain why those 2 isolates were chosen.

Reply: Indeed, this issue should clarify more. In our previous work done in 2022 (Kalman et al., Biology 2020, 9(4),69;

https://doi.org/10.3390/biology9040069), four Fusarium spp. Isolates were identified in diseased onion bulb samples collected from the basal rot (FBR) disease-contaminated field: F. proliferatum, F. oxysporum f. sp. cepae, F. acutatum, and F. anthophilium. Two of them, F. oxysporum f. sp. cepae and F. acutatum, were more prevalent in the field sample. Therefore, they were chosen for a subsequent work that tested chemical control interphase to restrain the disease (Degani and Kalman, J. Fungi 2021, 7(3), 235; https://doi.org/10.3390/jof7030235). This pioneering work against the north Israel FBR causal agents included plates’ screening of fungicides, in vitro seed pathogenicity assay, and potted seedlings in a growth room (we already elaborate on this in the manuscript, see lines 113-124). Still, the chemical interphase in the previous work didn’t reach a full season validation. So the current follow-up work aimed to identify new FBR protective compounds and to continue testing those already past the preliminary exams (up to sprouts level) in a semi-commercial full-season scale.

The following explanation was added to the text (lines 140-143): “These two isolates were more prevalent in field samples of basal rot diseased onion bulbs [2]. Therefore they were chosen for our previous chemical control study [15] and the current follow-up work.”

Line 132 - why in  Table 1 are there 12 fungicides but in Figure 1 only 9? Why Sportak, Azimut, and Vibrance were not tested in this experiment? So what is the Figure 1 for?

Reply: As explained above and in the text (lines 137-140), The current study aims to identify new highly effective FBR anti-fungal preparations and inspect those previously tested in sprouts over an entire growing season. Thus, nine new compounds were fast-screened in a plate assay against the major fungal agents of the disease. At the same time, three compounds that already passed the plate assy, seeds’ assay (to identify possible phytotoxicity), and sprout assay (pots in a controlled growth room), were evaluated for the first time in a semi-field whole season trial. Thus the total number of fungicides tested here is 12.

To clarify this issue, we added a new column to Table 1 to point out the fungicides used in each experiment. Also, the following explanation was added to Table 1 footnotes: “2 The fungicides tested in plates assay were evaluated, for the first time in Israel, against the onion basal rot disease (FBR) pathogens, F. oxysporum f. sp. cepae (B14 strain) and F. acutatum (B5 strain). Three pesticides (Sportak, Azimut, and Vibrance) were previously tested in plates, seeds, and sprouts assays and evaluated here in a full growing season.”

Indeed the new potential anti-fungal chemicals that had an encouraging score in the plates assay conducted here should be thoroughly examined in seeds, sprouts, and mature plants, in subsequent work. This aspect was already highlighted in the Discussion section (lines 538-546).

Line 132 - please explain whether you make the concentration from 99% AI or from the fungicide?

Reply: Thank you for this important remark. Indeed, we used commercial fungicides and not purified active ingredients.

This explanation was added to the text (lines 154-157): “Petri dishes (9 mm) containing potato dextrose agar medium (PDA; Difco Laboratories, Detroit, MI, USA) were prepared in which different commercial pesticides (Table 1) were combined with an active substance concentration of 1, 10 and 100 parts per million (ppm).”

Line 147 - Is it from treatment 2.1 or from the literature? How do you measure it as effective?

Reply: These three compounds were already tested by us in plates, seeds, and sprout assays (Degani and Kalman, J. Fungi 2021, 7(3), 235; https://doi.org/10.3390/jof7030235). So yes, this information is from the literature. Effectiveness or a high score in the plate assay is where a fungicide achieves a high statistical difference from the control (p << 0.05), even at the lowest dosage tested.

It was already written in the text that: “To examine the effect of chemical FBR control in pots, we used substances that were previously [15] found to be effective in the plates inhibition test…” (lines 179-180)

The following explanation was added to the text (lines 182-183): “ An effective compound had a high statistical difference from the control (p << 0.05), even at the lowest dosage tested”.

Line 183 - wheat seeds

Reply: Corrected to "wheat seeds" as suggested.

Line 184 - mixed Fusarium or Fusarium spp.? or separated for each species?

Reply: Separate infected grains batches were prepared for each species (with 10 colony agar disks from a single culture). We also set up a mixed Fusarium-infected wheat grains batch (with 5 colony agar disks from each of the two species cultures).

The above explanation was added to the text (lines 219-222).

Line 188 - a

Reply: Corrected to a small letter as advised.

Line 211 - How to measure? Why not measure the disease intensity? Because somehow plants look to survive but actually have been infected and could not be harvested?

Reply: Seedlings and dormant- or post-harvest bulbs are the most susceptible to FBR. Indeed the disease is sometimes referred to “damping-off” or “dieback” disease of the seedlings, i.e., the disease causes seedling mortality (see: le at al., Tropical Plant Pathology (2021) 46:241–253, https://doi.org/10.1007/s40858-021-00421-9. Thus surviving rate is an important measure to evaluate damping-off.

Still, FBR diseases can occur at all stages of Allium development. Symptoms become more conspicuous during crop maturity. The first signs of mature plants being infected are the yellowing of leaves, followed by symptoms of withered and curly leaves. Subsequently, these symptoms begin to spread downward. Rotting of bulbs and the appearance of root abscission layer are noticeable symptoms of infection, leading to easily uprooting bulbs from roots.

Thus, dehydration and delay development (disease severity) can be estimated by measuring the shoot’s fresh weight and height, water content, and the number of leaves (phenological progression), as done here (lines 246-249).

The following paragraph was added to the Introduction (lines 57-64): “Seedlings and dormant- or post-harvest bulbs are the most susceptible to FBR. Indeed the disease is sometimes referred to as the “dieback” or “damping-off” disease of the seedlings, i.e., the disease causes seedling mortality [8]. Still, FBR diseases can occur at all the Allium development stages. Symptoms become more noticeable during the plants’ maturity. The first symptom of plants being infected is the yellowing of leaves, followed by symptoms of curly and withered leaves. Subsequently, these symptoms begin to spread downward. Rotting of bulbs and the root abscission layer appearance are apparent signs of infection, leading to easily uprooting the plants and bulbs from roots [2].”

Line 272 - how about the dose? The low and high dose was not applied in this experiment? Please elaborate in the methods. I just read the low and high doses of Prochloraz and Az+Te

Reply: Thank you for this important remark. We actually made three seedling survival and emergence evaluations, 3, 14, and 35, after the beginning of the soil surface appearance. Those evaluations are already presented in Table 2. Still, we decided to include only the day 35 assessment results in the first manuscript draft since the results in this evaluation had more significant differences between the treatments. The first chemical treatment was performed 16 days after the beginning of the soil surface appearance, so the first two emergence evaluations were performed before this treatment, and the third was done afterward. Still, in each onion cultivar, the three soil surface peek evaluations (done in the same way, without referring to the high and low pesticide concentrations applied before the third measurement) gave similar results. The only difference was an overall decrease in the sprouts’ numbers and more identifiable statistical differences between the treatments in the third evaluation.

We agree with the reviewer that this information should be presented and explained. We, therefore, replaced Figure 3. The new figures now include all three sampling dates’ results. Accordingly, the figure legend was updated, and the following explanation was added to the text (lines 285-298):

“Three seedling survival and emergence evaluations were made, 3, 14, and 35, after the beginning of the soil surface appearance (Figure 3). The first chemical treatment was performed 16 days after the soil surface peeks started, so the first two emergence evaluations were performed before this treatment, and the third was done afterward. Still, in each onion cultivar, the three soil surface peek evaluations (done in the same way, without referring to the high and low pesticide concentrations applied before the third measurement) gave similar results. The only difference was an overall decrease in the sprouts’ numbers and more identifiable statistical differences between the treatments in the third evaluation. Is this mortality increase a consequence of the chemical first treatment’s phytotoxicity? Apparently not. The general lower survival parameters after 35 days were also measured in the free chemical control and were evidenced between the two first evaluations made before the chemicals were added. For example, the F. acutatum (B5) control peek values were decreased from ca. 60% (day 3) to 40% (day 14) and 30 % (day 35). Similarly, the F. oxysporum f. sp. cepae (B14) control values dropped from 75% to 60% and 40%.”

Line 333 - Prochloraz

Reply: Corrected to Prochloraz. Thank you.

Line 350 - water content?

Reply: This is a correct remark. We apologize for this mistake. Corrected to “water content.”

Line 358 - water content?

Reply: Corrected to “water content.”

Line 362 - I did not see the discussions about Fig 6. and 7. And what implies in water content result? Please also elaborate.

Reply: We agree; this aspect should explain and discuss more. So, the following paragraph was added to the Discussion section (lines 482-488): “The different growth measures used in the current study showed similar tendencies (for instance, see Figures 6 and 7). This is not surprising because the main FBR symptom at the maturity phase is dehydration of the plant’s upper parts (including the leaves and flowers) [8]. As a result of the water content drop, the plants lose weight, their development is disrupted (they are shorter and have fewer leaves), and their bulb development is delayed. Successful chemical protective treatments such as Prochloraz (Sportak) and Fl+Se (Vibrance) are reflected in recovering these indices, as demonstrated here.”

Line 405 - Please also elaborate why in some parameters Prochloraz were not significant different from control?

Reply: Prochloraz was dominant as a successful treatment in controlling pathogen F. oxysporum f. sp. cepae (B14) was well documented during the growth period up to 65 days from the surface peeking, in both onion cultivars tested. Yet at the season end (day 115), the treatment’s impact was more pronounced in the bulbs’ fresh weight and less in protecting the shoot. Most plants’ upper parts at this growth stage suffer from dehydration (shoot wet biomass reduction), and exclusively the Fl+Se (Vibrance) treatment could protect them. The difference between the two fungicides may be their residuals in plant tissues after the third application (day 56). Therefore, a preferred choice may be alternating Prochlorase with Vibrance towards the season-ending. Such a solution should be tested in future works.

The above explanation was added to the text (lines 452-461)

Line 471 - - ?

Reply: We removed the dash “-“ mark.

Reviewer 2 Report

This manuscript reports the control effect against Fusarium oxysporum f. sp. cepae and Fusarium acutatum of nine commercial fungicides. on the whole, several fungicides were found to be highly effective against the two pathogens, and three of these preparations were evaluated here at high and low dosages in a full growing season. This work is a useful contribution to the chemical control of Fusarium and I recommend acceptance, subject to the following changes:

1.     What are the technical difficulties in controlling Fusarium? Why constantly develop new preparations and combinations of preparations to control Fusarium? Suggest adding detailed information about these two questions in Introduction.

2.     In the fungicide culture plate assay, is there any reference for setting the activity concentration? Suggest supplement.

3.     Impact of the treatments at the season end (day 115): “The Az+Te treatment, even at the lowest dosage, appears phytotoxic. Thus, it will be necessary to reduce the dose of the preparation to evaluate its protective effect better.” What is the lowest dosage? Is there a basis for setting this lowest dosage? Does it represent the lowest dose that can prevent and control the disease? If the study continues to reduce the dose in order to better evaluate the protective effect, can the control effect be guaranteed?

Author Response

Responses to Reviewer 2’s comments

We would like to express our sincere appreciation to the reviewer for the essential and helpful advice. The time and effort invested are greatly appreciated and certainly contributed to the manuscript and improved it. Thank you. All your remarks and suggestions were addressed carefully and thoroughly, as detailed below.

This manuscript reports the control effect against Fusarium oxysporum f. sp. cepae and Fusarium acutatum of nine commercial fungicides. On the whole, several fungicides were found to be highly effective against the two pathogens, and three of these preparations were evaluated here at high and low dosages in a full growing season. This work is a useful contribution to the chemical control of Fusarium and I recommend acceptance, subject to the following changes:

Reply: Thank you for the positive evaluation of our manuscript. All your remarks and suggestions were addressed carefully and thoroughly, as detailed below.

  1. What are the technical difficulties in controlling Fusarium? Why constantly develop new preparations and combinations of preparations to control Fusarium? Suggest adding detailed information about these two questions in Introduction.

Reply: Thank you for this suggestion. Indeed, these are intriguing questions. The following explanation was added to the Introduction (lines 100-112):

“Successful chemical control methods can lose effectiveness or be unfitted to other areas. One central factor is the evolution of fungicide resistance that accelerates when extensive fungicides are applied in the same field for a long duration [13]. Another influencing factor is the variations in the Fusarium population composition. Thus, the efficacy of chemicals in controlling FBR in the field greatly varies and may not provide sufficient disease protection [14]. Although single Fusarium species could cause FBR, a complex of Fusarium species is frequently found to be responsible for the disease. Indeed, as many as 14 different Fusarium species were identified as causal agents of FBR around the globe [8]. Each of these species may respond differently to the chemical treatment, as proved by us lately [15]. Thus, searching for and developing new chemical-based treatments with new compounds is a continued effort with high priority. Essentially, to overcome the potential risk that the fungus will become resistant to fungicides, incorporating two or more active ingredients with a different mode of operation is necessary for long-term use.”

These aspects are also debated in detail in the Introduction (lines 114-137) and Discussion (see lines 547-554).

  1. In the fungicide culture plate assay, is there any reference for setting the activity concentration? Suggest supplement.

Reply: We agree. The following explanation was added to Materials and Methods (lines 158-163):

“The fungicide concentration range tested (1-10 ppm) in the plate assy is quite wide. The minimum of 1 ppm concentration often has minor to no impact on the fungal growth in many anti-fungal preparations. On the other side, the high 100 ppm dosage can drastically reduce the fungal colony growth in successful pesticides. Thus it is an adequate concentration range for these kinds of tests. A similar dosage range was previously used by us ([15,18]) and by others ([16,19]) against plant fungal pathogens.”

  1. Impact of the treatments at the season end (day 115): “The Az+Te treatment, even at the lowest dosage, appears phytotoxic. Thus, it will be necessary to reduce the dose of the preparation to evaluate its protective effect better.”What is the lowest dosage? Is there a basis for setting this lowest dosage? Does it represent the lowest dose that can prevent and control the disease? If the study continues to reduce the dose in order to better evaluate the protective effect, can the control effect be guaranteed?

Reply: These are indeed excellent questions; thank you. To the best of our knowledge, the Azoxystrobin 12% + Tebuconazole 20% mixture (Az+Te, Azimut) treatment was evaluated here for the first time against the Israel FBR causal agents. Therefore, the lowest effective dosage of this pesticide is yet to be determined. The phytotoxic effect of anti-fungal compounds on the cultivated plants varied depending on the dose, the application method and timing, and the test cultivar. So a delicate adjustment should be tailored in each chemo-control protocol to achieve maximum efficiency and minimum plant toxicity. Such a task becomes more complex and challenging when a few fungal pathogens species involved with each have a different response to the pesticide (as demonstrated here). It is a challenge to subsequent works to fine-tune the control interphase and maximize its potential. Such future interphase will probably rely on ingredients with a different active mechanism (to avoid resistance). So, using compounds’ mixtures or applying various pesticides in a sequence must be considered.

The following paragraph was added to the discussion (lines 490-497): ”The Azoxystrobin 12% + Tebuconazole 20% mixture (Az+Te, Azimut) treatment was probably phytotoxic and damaged the plants’ development even at the lowest dosage. The phytotoxic effect of anti-fungal compounds on the cultivated plants varied depending on the dose, the application method and timing, and the test cultivar. So a delicate adjustment should be tailored in each chemo-control protocol to achieve maximum efficiency and minimum plant toxicity. Such a task becomes more complex and challenging when a few fungal pathogens species involved with each have a different response to the pesticide (as demonstrated here).”

Round 2

Reviewer 1 Report

I am happy with the revised manuscript and accept the revision for publication. 

Author Response

Thank you for the essential and constructive advice. The time and effort you invested are greatly appreciated and certainly contributed to the manuscript and improved it.